# The Reprimo-Like Gene Is an Epigenetic-Mediated Tumor Suppressor and a Candidate Biomarker for the Non-Invasive Detection of Gastric Cancer

**DOI:** 10.3390/ijms21249472

**Published:** 2020-12-12

**Authors:** María Alejandra Alarcón, Wilda Olivares, Miguel Córdova-Delgado, Matías Muñoz-Medel, Tomas de Mayo, Gonzalo Carrasco-Aviño, Ignacio Wichmann, Natalia Landeros, Julio Amigo, Enrique Norero, Franz Villarroel-Espíndola, Arnoldo Riquelme, Marcelo Garrido, Gareth I. Owen, Alejandro H. Corvalán

**Affiliations:** 1Department of Hematology & Oncology, Pontificia Universidad Católica de Chile, Santiago 8330034, Chile; mralarco@uc.cl (M.A.A.); wdolivar@uc.cl (W.O.); cordova.delgado.m@gmail.com (M.C.-D.); matiasm.m@outlook.com (M.M.-M.); wichmann@uc.cl (I.W.); natalialanderos@udec.cl (N.L.); mgarrido@med.puc.cl (M.G.); 2Advanced Center for Chronic Diseases (ACCDiS), Pontificia Universidad Católica de Chile, Santiago 8330034, Chile; tomas.demayo@mayor.cl (T.d.M.); gowen@bio.puc.cl (G.I.O.); 3Faculty of Sciences, School of Medicine Universidad Mayor, Santiago 8580745, Chile; 4Department of Pathology, Hospital Clínico Universidad de Chile, Santiago 8380456, Chile; gcarrasco@clinicalascondes.cl; 5Department of Pathology, Clínica Las Condes, Santiago 7591210, Chile; 6Department of Obstetrics, Pontificia Universidad Católica de Chile, Santiago 8330024, Chile; 7Department of Physiology, Pontificia Universidad Católica de Chile, Santiago 8330005, Chile; jamigo@bio.puc.cl; 8Esophagogastric Surgery Unit, Hospital Dr Sótero del Río, Santiago 8207257, Chile; enorero@uc.cl; 9Digestive Surgery Department, Pontificia Universidad Católica de Chile, Santiago 8330024, Chile; 10Translational Medicine Laboratory, Instituto Oncológico Fundación Arturo López Pérez (FALP), Santiago 8320000, Chile; franz.villarroel@falp.org; 11Department of Gastroenterology, Pontificia Universidad Católica de Chile, Santiago 8330024, Chile; a.riquelme.perez@gmail.com

**Keywords:** biomarkers, gastric cancer, plasma, non-invasive, methylation, intronless

## Abstract

Reprimo-like (*RPRML*) is an uncharacterized member of the Reprimo gene family. Here, we evaluated the role of *RPRML* and whether its regulation by DNA methylation is a potential non-invasive biomarker of gastric cancer. RPRML expression was evaluated by immunohistochemistry in 90 patients with gastric cancer and associated with clinicopathologic characteristics and outcomes. The role of *RPRML* in cancer biology was investigated in vitro, through *RPRML* ectopic overexpression. Functional experiments included colony formation, soft agar, MTS, and Ki67 immunofluorescence assays. DNA methylation-mediated silencing was evaluated by the 5-azacytidine assay and direct bisulfite sequencing. Non-invasive detection of circulating methylated *RPRML* DNA was assessed in 25 gastric cancer cases and 25 age- and sex-balanced cancer-free controls by the MethyLight assay. Downregulation of RPRML protein expression was associated with poor overall survival in advanced gastric cancer. *RPRML* overexpression significantly inhibited clonogenic capacity, anchorage-independent growth, and proliferation in vitro. Circulating methylated *RPRML* DNA distinguished patients with gastric cancer from controls with an area under the curve of 0.726. The in vitro overexpression results and the poor patient survival associated with lower RPRML levels suggest that RPRML plays a tumor-suppressive role in the stomach. Circulating methylated *RPRML* DNA may serve as a biomarker for the non-invasive detection of gastric cancer.

## 1. Introduction

Gastric cancer (GC) remains the third leading cause of cancer-related death globally [1]. However, its mortality depends greatly on the stage in which it is diagnosed. Unfortunately, the standard diagnostic method, esophagogastroduodenoscopy, is an invasive and expensive procedure which has resulted in late diagnosis and an average 5-year survival of ~30%. Therefore, elucidating the molecular basis of GC has become crucial to developing timely diagnostic and therapeutic strategies [2,3].

Comprehensive studies have revealed that extensive epigenetic DNA hypermethylation on tumor suppressor genes (TSG) is a recurrent phenomenon in GC [4]. These aberrant patterns of DNA methylation arise early during carcinogenesis and, when occurring within the nearest region surrounding the transcription start site (TSS) (± 200 bp) may lead to TSG silencing [5,6]. These genes are involved in crucial tumorigenic processes such as cell cycle regulation (*p16NK4a*, *p15INK4b*, *p14ARF*, *RPRM*, *RUNX3*), apoptosis (*XIAP*, *BCL2*, *DAPK*, *BNIP3*, *TMS1*, *CASP8*, *GPX3*, *RNF180*), DNA repair (*hMLH1*, *MSH2*, *MGMT*), migration, and invasion (*CDH1*, *CDH4*, *PCDH10*, *RASSF1A*, *ZIC1*, *APC*, *GRIK2*, *FLNC*, *LOX*, *TIMP3*, *TSP1*), among others, and have been associated with the progression and poor prognosis of GC [7,8]. Moreover, it has been demonstrated that circulating tumor-specific methylated DNA can be detected in different biological fluids, as well as microRNAs, long non-coding RNAs, extracellular vesicles, and circulating tumor cells (CTCs) [9]. Thus, epigenetic-mediated TSG silencing not only contributes to the pathogenesis of GC but has also been proposed as a candidate biomarker for the non-invasive diagnosis and monitoring of disease [10,11,12]. 

The Reprimo (*RPRM*) gene family is a poorly characterized set of intronless genes with expression patterns that have been recently associated with gastrointestinal tract development [13]. This gene family originated early during vertebrate evolution, with two of its members conserved in humans: *RPRM* and *RPRM*-like (*RPRML*) [14]. *RPRM*, the founding member of this family, is a TSG involved in cell cycle control downstream of p53 [15,16,17] and is silenced by DNA methylation in human tumors, which has been explored as a non-invasive biomarker in GC [17,18,19].

The second member of this gene family, *RPRML*, is also an intronless gene expressed at very low levels in most tissues according to the Genotype-Tissue Expression (GTEx) database (www.gtexportal.org). Despite their low level of expression, these genes encode proteins with important biological functions [20]. In the present study, we evaluated the functional properties, clinical significance, and potential translational applications of *RPRML*, a hitherto uncharacterized member of the *RPRM* gene family in GC.

## 2. Results

### 2.1. RPRML Expression in Clinical Samples

We explored RPRML protein expression in stomach tissues through an immunohistochemical (IHC) staining assay. This analysis was performed in 14 normal gastric mucosa tissues and 17 matched pairs of gastric tumors and non-tumor adjacent mucosa (NTAM). Both in normal gastric mucosa and in NTAM, weak to moderate cytoplasmic RPRML protein expression was seen in glandular and foveolar epithelial cells (Appendix A and Figure 1a). The matched tumor tissues showed weak and heterogenous cytoplasmic staining. Semiquantitative scoring of matched NTAM and tumor samples resulted in a median RPRML IHC score of 1.5 (interquartile range (IQR): 1.0–1.5) and 0.5 (IQR: 0.1–1.0), respectively (Figure 1b). Paired sample analysis revealed a significant downregulation in the tumor tissues (*p* = 0.001) (Figure 1b). These findings were corroborated by in silico analysis of RNAseq expression data from 32 matched pairs of tumor tissues and NTAM from The Cancer Genome Atlas Stomach Adenocarcinomas (TCGA-STAD) dataset [21], showing consistent downregulation of RPRML mRNA in tumor tissues (*p* = 0.01) (Appendix A).

To evaluate the clinical significance of the loss of RPRML, IHC staining was assessed in a cohort of 90 patients with GC [22]. Using the RPRML IHC score as a continuous variable, clinicopathological features such as age, sex, Lauren histological classification, tumor localization, and tumor–node–metastasis (TNM) stage showed no statistically significant differences (Appendix A). Low RPRML IHC scores were associated exclusively with low cleaved (Cl) caspase-3, among several tissue markers (Appendix A). Additionally, the multivariate Cox model adjusted by sex, age, and TNM stage, showed that low RPRML expression was significantly associated with worse overall survival (OS) (Appendix A). Further analysis according to TNM stage subgroup showed that low RPRML expression was a significant risk factor for patients with advanced GC (Hazard Ratio (HR) 0.07, 95% confidence interval (CI): 0.01–0.46, *p* = 0.005) (Table 1).

To gauge the impact of RPRML expression on OS, all advanced GC cases were stratified into high- and low-RPRML expression groups using an optimal cut-off value for RPRML IHC score previously determined by receiver operating curve (ROC) curve analysis (Appendix A). With this approach, the 2- and 5-year survival rates in the low-expression group were less than half those of the high-expression group (2-year survival = 40.0 vs. 81.3 months; 5-year survival = 17.0 vs. 53.5 months, respectively) (Figure 2). The overall comparison showed that the low-expression group had significantly worse prognosis compared with the high-expression group (*p* = 0.00051, log-rank test). Notably, the low-expression group had a median OS of 16 months, while patients with high RPRML expression did not reach the median OS. Taken together, these results indicate that *RPRML* downregulation is a risk factor for poor prognosis in advanced-stage GC.

### 2.2. Regulatory Mechanisms of RPRML Expression in GC

To explore the potential mechanisms mediating RPRML silencing in GC, germline and somatic genetic alterations were evaluated by sequencing a cohort of 36 patients with familial GC and retrieving data from 393 sporadic cases from the TCGA-STAD dataset [21,23]. These analyses resulted in two likely benign RPRML germ cell variants in the familial GC cohort (Appendix A), and no somatic inactivating mutations in the sporadic TCGA dataset, respectively. Taken together, these results suggest that genetic alterations would not be a significant cause of the inactivation of the RPRML gene.

The RPRML gene is located within a dense CpG island in the genome [24], suggesting that DNA methylation may mediate RPRML silencing in GC (Figure 3a). Therefore, we evaluated RPRML promoter methylation status in three GC cell lines (AGS, Hs746T, SNU-16) with undetectable RPRML transcript expression (Appendix A). The analysis showed that all CpG sites within the +71 to +289 region relative to the TSS [25] were methylated in the three evaluated cell lines (Figure 3b). Treatment of SNU-16 cells with 1 µM DNA methylation inhibitor 5-azacytidine (5-Aza) resulted in CpG demethylation (Figure 3b, bottom panel). To confirm that RPRML silencing was mediated by DNA methylation, we assessed re-expression of RPRML after 5-Aza treatment in the above cell lines. Figure 3c shows that treatment with 1 µM 5-Aza restored RPRML transcription in the Hs746T and SNU-16 cell lines but not in the AGS cell line. RPRML transcription was not restored upon increasing the concentration of 5-Aza to 5 µM (data not shown). These results suggest that DNA methylation plays a significant role in regulating RPRML transcription; however, as in the case of the AGS cell line, additional mechanisms may be involved.

### 2.3. In Vitro Characterization of RPRML Functionality

Due to the homology with the founding member of the RPRM family, we evaluated if RPRML also possessed tumor suppressor properties. To this end, the AGS primary GC cell line (that lack RPRML transcript expression; Appendix A) was stably transfected with GFP-tagged RPRML or GFP alone. Transfected cells were recovered by fluorescence-activated cell sorting (FACS) and expression was confirmed by fluorescence microscopy and Western blotting (Appendix A). Figure 4 shows that cells with RPRML overexpression significantly reduced AGS cell clonogenic capacity and anchorage-independent growth, suggesting a tumor suppressor function in vitro. The MTS assay revealed that RPRML overexpression significantly reduced cell proliferation at 24 h, 48 h, and 72 h after seeding (Figure 4e). Furthermore, analysis of the cell proliferation protein Ki67 by immunofluorescence confirmed a significant reduction in the presence of RPRML in comparison with both wild type (WT) and control GFP-expressing cells (*p* < 0.05) (Figure 4f,g). Cell cycle progression analysis suggested that this reduction in proliferation may be due to an arrest in G2/M (Appendix A). Interestingly, X-ray–induced DNA damage did not increase RPRML expression (Appendix A). Taken together, these results suggest that RPRML reduces cell proliferation, supporting its role as a tumor suppressor in GC.

### 2.4. Circulating Methylated RPRML DNA in Plasma Samples for the Non-Invasive Detection of GC

As our earlier results suggested that RPRML expression is consistently downregulated in GC and that this is mediated by DNA methylation, we explored the role of circulating methylated RPRML DNA as a non-invasive biomarker for detecting GC. To this end, we developed a MethyLight assay covering 10 CpGs from a 142-bp target region near the TSS of the RPRML gene. Methylated RPRML DNA was quantified in plasma samples from 50 patients: 25 GC cases, and 25 cancer-free controls. The results were analyzed by ROC curve analysis, yielding an area under the curve (AUC) of 0.726 (95% CI: 0.583–0.869, *p* = 0.006) (Figure 5). The cut-off point that maximized the sensitivity and specificity for detecting GC was 1.0 copy/mL plasma. Using this cut-off value, we detected circulating methylated RPRML DNA in 14 of the 25 GC patients, yielding a sensitivity of 56.0% (95% CI: 34.93–75.60). Three of the 25 controls were positive for methylated RPRML DNA, yielding a specificity of 88.0% (95% CI: 68.78–97.45). The positive likelihood ratio (LR+) was 4.67 (95% CI: 1.53–14.26) and the negative likelihood ratio (LR-) was 0.50 (95% CI: 0.31–0.80). The odds ratio was 9.34 (95% CI: 2.20–39.46, *p* = 0.002). These results suggest that circulating methylated RPRML DNA may be useful for non-invasive diagnosis of GC.

## 3. Discussion

Early detection offers the opportunity for increasing the survival rates of patients with GC. Aberrant DNA methylation occurs early in the course of gastric carcinogenesis and is recognized as a promising biomarker for non-invasive cancer detection [26,27]. In the present study, we found that detecting circulating methylated *RPRML* DNA in plasma samples significantly distinguished patients with GC from cancer-free controls (AUC: 0.726, *p* = 0.006). 

A recent meta-analysis compared the clinical performance of methylation-based blood biomarkers for detecting GC [28]. Among the biomarkers with significant discriminatory capacity, *RPRM,* the homolog of *RPRML,* was proposed as one of the most promising candidates. Our result of the odds ratio of *RPRML* (9.34, 95% CI: 2.20–39.46) falls among the mean odds ratios of these reported biomarkers, which ranged from 3.16 (95% CI: 1.47–6.81) for *MGMT* to 111.1 (95% CI: 36.67–336.59) for *RPRM* [28]. However, a wide 95% CI was observed in each of these candidates, indicating low-precision accuracy and inconsistency between aggregated studies. 

Due to the heterogeneous nature of GC, it is highly unlikely that the use of a single biomarker will achieve sufficient sensitivity for screening purposes [6]. Recent analyses have clearly shown the superiority of multi-biomarker panel approaches for detecting GC; however, novel individual candidates are still needed to improve reliability [29,30,31]. Thus, circulating methylated *RPRML* DNA may contribute to increasing the sensitivity of a multi-biomarker panel without adding considerable false positives to the test.

Aberrant DNA methylation may lead to the transcriptional silencing of tumor-related genes and thus contribute to the pathogenesis of cancer [32]. In the present study, 5-Aza demethylation treatment restored *RPRML* transcript expression in two GC cell lines with prior undetectable *RPRML* mRNA (SNU-16 and Hs746T). Moreover, analysis of the DNA methylation pattern near the TSS [25] confirmed that *RPRML* transcript expression was regulated by DNA methylation. Conversely, 5-Aza treatment did not restore *RPRML* expression in the AGS cell line in the absence of mutational inactivation [33]. As observed with other TSGs, our results suggest that additional layers of transcriptional regulation may restrict *RPRML* expression [34].

Herein, we also provide evidence that *RPRML* has tumor-suppressive properties. Overexpression of *RPRML* in the AGS cell line significantly inhibited clonogenic capacity and anchorage-independent growth. Moreover, *RPRML* overexpression reduced AGS cell proliferation by arresting the cell cycle at the G2/M phase. These observations support a tumor-suppressive role of *RPRML* and suggest a cell cycle-related function, similar to its homolog *RPRM* [14,17,20]. Correspondingly, analysis of RPRML expression in NTAM and tumor tissues showed consistent downregulation in clinical samples and was associated with worse prognosis in advanced stages of GC. These results suggest that the loss of RPRML protein expression is an independent prognostic factor and may drive GC progression. Thus, it provides the opportunity for exploring the potential of RPRML as an actionable target for advanced GC, as has been previously proposed for its homolog *RPRM* using Clustered Regularly Interspaced Short Palindromic Repeats (CRISPR) technology [35]. Interestingly, the absence of RPRM gene expression in clinical samples is not associated with poor prognosis in GC patients [15]. However, the apoptosis-resistant phenotype mediated by survivin, a member of the inhibitor-of-apoptosis protein (IAP) family, has been reported to be detrimental to survival in GC only in the absence of RPRM gene expression [15]. In addition, the loss of RPRML protein expression in clinical samples was associated with reduced Cl-caspase-3 immunostaining. This finding suggests that RPRML may have a role in resisting apoptosis, which has been previously associated with poor GC prognosis [36]. 

Our findings are subject to certain limitations. It should be noted that due to the exploratory nature of this study, a small case–control design was used to evaluate the potential of circulating methylated *RPRML* DNA as a non-invasive biomarker of GC. We did not consider variables such as *Helicobacter pylori* and Epstein–Barr virus infections, which have been reported to influence the methylation status in the stomach [37]. In addition, the diagnostic accuracy of circulating methylated *RPRML* DNA and the prognostic value of RPRML protein expression warrant validation in independent cohorts. Finally, further functional analyses and animal studies will be required to fully validate the role of *RPRML* and its specific signaling pathways in GC.

Despite these considerations, this study constitutes a first step toward a broader characterization of this hitherto undescribed gene in human biology and pathology. Our results suggest that *RPRML* is a TSG, downregulated by DNA methylation in GC, and that circulating methylated *RPRML* DNA can distinguish patients with GC from cancer-free controls. Thus, these findings provide justification for larger clinical studies to further assess its value in multi-biomarker panel approaches for non-invasive diagnosis of GC.

## 4. Materials and Methods

### 4.1. Clinical Samples and Pathological and Follow-Up Data

Formalin-fixed and paraffin-embedded (FFPE) stomach tissue samples from 14 de-identified cancer-free controls who were recruited between 2010 and 2012 from an upper gastrointestinal endoscopic screening program at Centro de Referencia de Salud La Florida were included. FFPE whole-tissue sections from 17 de-identified patients with GC who had undergone total gastrectomy between 2008 and 2012 were retrospectively collected from the archives of the Pathology Department of Hospital Clínico Universidad de Chile (HCUCH). Tissue microarrays (TMAs) including 90 GC cases enrolled between 2004 and 2018 at Centro de Cancer UC-CHRISTUS, Pontificia Universidad Católica de Chile (PUC), together with anonymized demographic, pathological, and follow-up data, were obtained from the FORCE1 clinical trial (NCT03158571). A detailed description of this cohort study and TMA construction can be found in Cordova-Delgado et al. [22]. Thirty-six de-identified germline DNA samples from a prospective cross-sectional familial GC study conducted between 2016 and 2020 at PUC were obtained for genetic screening of the *RPRML* gene. A detailed description of this cohort can be found in Norero et al. [23]. Plasma samples from 25 GC cases and 25 age- and sex-balanced cancer-free controls were prospectively collected at Hospital Clinico Universidad Católica UC, Biobanco de Tejidos y Fluidos Universidad de Chile (BTUCH), and Fundación Arturo Lopez Perez. The median age of cases was 62.0 (IQR: 54.0–67.0) and the median age of controls was 59.0 (IQR: 54.0–66.0). The male to female ratio in both groups was 2.1:1. The clinical diagnosis of cancer-free controls and GC was obtained from anonymized pathology reports. Cancer-free patients were defined by the Operative Link of Gastritis Assessment (OLGA) staging system as OLGA 0, I or II [38]. All samples and data were used in accordance with the principles of the Helsinki declaration. Ethical approval was obtained from the Internal Review Board and the Ethics and Scientific Committee at PUC—School of Medicine (Protocol #10-061, 19 August 2010; FORCE1 #16-046, 21 April 2016; Protocol #180822037, 2 May 2019; and FONDECYT #1151411, 5 June 2018). Written informed consent forms were obtained from all participants and a consent waiver was granted in the case of deceased patients. 

### 4.2. Immunohistochemical Analysis

RPRML immunohistochemistry was performed using a polyclonal anti-RPRML antibody (Abcam, Cambridge, UK, Cat# ab204896, RRID: AB_2861374) and VECTASTAIN^®^ ABC R.T.U universal kit (Vector Laboratories) according to the manufacturer’s instructions. Briefly, 4-µm FFPE TMA or tissue sections were deparaffinized and rehydrated through xylene and a graded alcohol series. Antigen retrieval was performed in an EDTA buffer at pH 9 (Agilent, Santa Clara, CA, USA) for 20 min. Endogenous peroxidase activity was blocked with a 4% hydrogen peroxide solution in methanol. Non-specific protein binding was blocked with VECTASTAIN^®^ normal horse serum (2.5%) for 10 min. RPRML immunostaining was performed using a 1:500 dilution in Emerald diluent (ESBE Scientific, Markham, ON, Canada) and 1-h incubation at room temperature. Slides were incubated for 12 min with VECTASTAIN^®^ biotinylated secondary universal antibody, followed by 12-min incubation with VECTASTAIN^®^ ABC reagent. Slides were developed with 3,3-diaminobenzidine substrate (Agilent, Santa Clara, CA, USA) for 1 min and counterstained with Meyer’s hematoxylin (ScyTek Laboratories, Logan, UT, USA). The slides were dehydrated and mounted with a synthetic hydrophobic resin (Thermo Fisher Scientific, Waltham, MA, USA). The IHC score was determined by calculating the product of a 4-point intensity score (0: no staining; 1: weak; 2: moderate; 3: strong), and the proportion of stained cells (range, 0–1) [39]. The specificity of the RPRML antibody was tested by Western blot analysis of ectopically overexpressed RPRML tagged with GFP (green fluorescent protein) (Appendix A). IHC of cleaved (Cl) caspase-3 was performed using the anti-Cl-caspase-3 antibody (1:2000, Cell Signaling Technology, Danvers, MA, USA, Cat# 9664, RRID: AB_2070042) as previously described [40]. The slides were examined by two pathologists blinded to the clinical data. The inter-observer interclass correlation coefficient (ICC) was 0.863 (95% confidence interval (CI): 0.789–0.911). Differences in interpretation were resolved by consensus. Scores from duplicate TMA cores and between the two pathologists were averaged.

### 4.3. Cell Culture and Transfections

The human GC cell lines AGS (ATCC, Cat# CRL-1739, RRID: CVCL_0139), Hs746T (ATCC, Cat# HTB-135, RRID: CVCL_0333), and SNU-16 (ATCC, Cat# CRL-5974, RRID: CVCL_0076) were purchased from ATCC. All cell lines were periodically tested using an EZ-PCR™ Mycoplasma Detection Kit (Biological Industries, Beit HaEmek, Israel) and the number of passages did not exceed 20. AGS and SNU-16 cells were cultured in HyClone RPMI-1640 medium (Cytiva, Washington, DC, USA), supplemented with 10% FBS (Biological Industries, Cromwell, CT, USA), 100 U/mL penicillin, and 100 µg/mL streptomycin. Hs746T cells were cultured in HyClone DMEM medium (Cytiva) supplemented with 20% FBS. Maintenance was performed at 37 °C in a humidified 5% CO_2_ incubator. AGS cells were transfected with a GFP-tagged human RPRML ORF (open reading frame) clone (pCMV6-RPRML-GFP, OriGene Technologies, Rockville, MD, USA, Cat# RG207364) or a GFP empty vector (pCMV6-AC-GFP, OriGene Technologies, Cat# PS100010) using FuGENE HD Transfection Reagent (Promega, Madison, WI, USA) according to the manufacturers’ protocol. Stable transfection was generated by antibiotic selection with 500 µg/mL G418 (Thermo Fisher Scientific, Waltham, MA, USA) for 21 days. Surviving cells were sorted by GFP fluorescence in a BD FACSAriaII cell sorter (BD Biosciences, San Jose, CA, USA, RRID: SCR_018934) and maintained in a culture medium containing 250 µg/mL G418.

### 4.4. Western Blot Analysis

Whole-cell lysates were extracted from 60-mm cell culture plates using RIPA buffer (Thermo Fisher Scientific, Waltham, MA, USA) containing Halt™ protease and a phosphatase inhibitor cocktail (Thermo Fisher Scientific). Total protein content was quantified using the Pierce BCA Protein Assay Kit (Thermo Fisher Scientific) following the manufacturer’s protocol. Equal amounts of proteins (20 µg) were separated on 12% SDS-PAGE and transferred to PVDF membranes. The membranes were blocked with 5% milk in TBS-T buffer for 1 h and incubated at 4 °C overnight with anti-RPRML antibody (Abcam, Cambridge, United Kingdom, Cat# ab204896, RRID: AB_2861374) (1:1000 in TBS-T/3% BSA), anti-turboGFP antibody (1:5000, Thermo Fisher Scientific, Cat# PA5-22688, RRID: AB_2540616), or anti-GAPDH antibody (1:2000, Cell Signaling Technology, Danvers, MA, USA, Cat# 97166, RRID: AB_2756824). Subsequently, the membranes were washed 5 times in TBS-T (5 min/wash) and incubated for 1 h at room temperature with the secondary anti-rabbit peroxidase-conjugated antibody (1:2000 in TBS-T) (Agilent, Santa Clara, CA, USA, Cat# P0448, RRID: AB_2617138). The membranes were washed and visualized using SuperSignal West Dura Chemiluminescent Substrate (Thermo Fisher Scientific, Waltham, MA, USA).

### 4.5. RNA Isolation and RT-PCR

Total mRNA was isolated from 100-mm culture plates using TRIzol (Thermo Fisher Scientific, Waltham, MA, USA) according to the manufacturer’s instructions. Reverse transcription of 1 µg mRNA was performed with Moloney Murine Leukemia Virus Reverse Transcriptase (M-MLV RT) (Promega, Madison, WI, USA) according to the manufacturer’s recommendations in a final volume of 20 µL. *RPRML* PCR amplification was performed using the following primers: forward, 5′-ATGAACGCGACCTTCCTGAAC-3′; reverse, 5′-GTGGGTGCGGTTTCCCA-3′. The reaction contained 2 µL cDNA, 2 mM MgCl_2_, 5 mM dNTPs, 0.25 µM of each primer, 5 µL 5X Green GoTaq^®^ Flexi Buffer, and 1 U GoTaq^®^ G2 Flexi DNA Polymerase (Promega, Madison, WI, USA) in a final volume of 25 µL. The thermal profile consisted of an initial 5-min incubation at 95 °C, followed by 35 cycles of denaturation, annealing, and extension periods at 95 °C, 60 °C, and 72 °C, respectively (30 s each) and a final extension of 5 min at 72 °C. The RT-PCR products were resolved on 2% agarose gels. *RPS13* was used as a reference gene.

### 4.6. DNA Isolation and Bisulfite Modification

Genomic DNA was isolated from confluent 100-mm culture plates using the Wizard SV Genomic DNA Purification System (Promega, Madison, WI, USA). Plasma DNA (500 µL) was isolated using the QIAamp DNA Blood Mini Kit (Qiagen, Hilden, Germany) following the manufacturer’s recommendations. Both isolations had a final elution volume of 100 µL. Genomic DNA (1 µg) or 20 µL plasma DNA underwent sodium bisulfite modification using an EZ DNA Methylation-Gold Kit (Zymo Research, Irvine, CA, USA) according to the manufacturer’s protocol, with a final elution volume of 20 µL.

### 4.7. Genetic Screening

Germline genetic screening of the entire coding sequence of the *RPRML* gene, including 200 bp from the proximal promoter and 178 bp from the 3′ untranslated region (UTR) was performed by Sanger sequencing. Briefly, genomic DNA was amplified using the following primers: forward1, 5′-CAGGAGGGGGTGGAGTTTCG-3′; reverse1, 5′-AGCCCGTGGGTGCGGTTTC-3′: forward2, 5′-GAACGCGACCTTCCTGAAC-3′; reverse2, 5′-GGCGTCCAAGTCCCTTC-3′. Amplification was performed using GoTaq^®^ G2 Flexi DNA Polymerase (Promega, Madison, WI, USA) according to the manufacturer’s instructions. The thermal profile was the same as for cDNA amplification. The PCR products were sequenced through the Macrogen sequencing service (RRID: SCR_014454). *RPRML* somatic mutations and copy number alterations data were retrieved from 393 patients from The Cancer Genome Atlas (TCGA)–Stomach Adenocarcinoma (STAD) dataset (Firehose Legacy) using the online cBioPortal tool (http://www.cbioportal.org, RRID: SCR_014555) accessed in August 2020. 

### 4.8. Direct Bisulfite Sequencing

The *RPRML* TSS (transcription start site) flanking region was amplified from bisulfite-modified genomic DNA using the following primers: forward, 5′-GGTGTTTAGGGGTAGG-3′; reverse, 5′-TCCACCTCCTCCAAAC-3′. The thermal profile was as described above with an annealing temperature of 55 °C. The PCR products were sequenced through the Macrogen service.

### 4.9. 5-Azacytidine Assay

AGS, Hs746T, and SNU-16 cells were treated with 5-azacytidine (5-Aza) compound (Abcam, Cambridge, United Kingdom, Cat# ab142744) as described by Bernal et al. [18]. Briefly, 1 × 10^6^ cells were seeded in 60-mm plates; 24 h later, the culture medium was supplemented with 1 or 5 µM 5-Aza. The 5-Aza–supplemented medium was changed every day for 3 days.

### 4.10. Colony Formation and Soft Agar Assays

Cells were seeded in 12-well plates (300 cells/well) and cultured for 14 days. Surviving colonies (>50 cells per colony) were counted under a light microscope after fixing and staining with 0.5% crystal violet in 25% methanol/1× PBS. Anchorage-independent cell growth was determined by a soft agar assay as described by Borowicz et al. [41]. Cells (5 × 10^3^/well) were mixed with 0.3% UltraPure™ LMP Agarose (Invitrogen, Carlsbad, CA, USA) in RPMI-1640 medium and plated on a solidified layer of 0.6% agarose in RPMI-1640–10% FBS medium in a 12-well plate. On Day 21, cells were fixed with cold 10% methanol in 1× PBS for 15 min and stained with 0.0001% crystal violet. Colonies > 50 µm in diameter were counted in each well under a light microscope.

### 4.11. MTS Assay

Cells were seeded on 96-well plates (5 × 10^3^ cells/well) and cultured for 0 h, 24 h, 48 h, and 72 h. At each time point, a 3-(4,5-dimethylthiazol-2-yl)-5-(3-carboxymethoxyphenyl)-2-(4-sulfophenyl)-2H-tetrazolium (MTS) assay (CellTiter 96^®^ AQ_ueous_ One Solution, Promega, Madison, WI, USA) was performed following the manufacturer’s instructions.

### 4.12. Ki67 Immunofluorescence

Cells (3 × 10^4^) were seeded on 12-mm cover slides and cultured for 48 h, then washed twice with 1× PBS and fixed with a buffered formalin-zinc solution (Thermo Fisher Scientific, Waltham, MA, USA) at room temperature for 30 min. The cells were permeabilized with 0.1% Triton X-100 in Tris-HCl for 15 min and incubated overnight with anti-Ki67 antibody (1:200, Abcam, Cambridge, United Kingdom, Cat# ab15580, RRID: AB_443209) in Emerald diluent (ESBE Scientific, Markham, ON, Canada). Secondary antibody Alexa Fluor 546 anti-rabbit IgG (H+L) (1:1000 dilution, Thermo Fisher Scientific, Cat# A-11035, RRID: AB_2534093) was incubated for 1 h at room temperature and Hoechst 33342 nucleic stain (1:2000, Thermo Fisher Scientific) for 15 min. Cells were visualized under an epifluorescence microscope (Zeiss, Oberkochen, Germany) and the AxioVision Imaging System (RRID: SCR_002677). The percentages of Ki67-positive cells were determined by counting five random fields at ×10 magnification using ImageJ v 1.52q (National Institutes of Health, Bethesda, Maryland, USA, RRID: SCR_003070).

### 4.13. MethyLight Assay

Bisulfite-modified plasma DNA was amplified by a MethyLight assay [42,43] using a Rotor-Gene Q 5plex Platform (Qiagen, Hilden, Germany). *RPRML* locus-specific amplification (+11 to +152 from the TSS) was performed using the following primers and fluorescent reporter probe: forward, 5′-TTCGGTTTTAGTTTTTGCGTC-3′; reverse, 5′-AACCGACTCCTACGATACGAA-3′; probe, 5′-FAM-CGGTTCGAGAGCGCGTAGGTAGTTA-TAMRA-3′. The MethyLight reaction was performed using 4 µL bisulfite-modified plasma DNA, 1× LightCycler FastStart DNA Master HybProbe (Roche, Basel, Germany), 0.6 µM each primer, and 0.2 µM oligonucleotide probe. The thermal profile was: 95 °C for 10 min, followed by 45 cycles at 95 °C for 5 s and 60 °C for 55 s. Amplification of a methylation-independent sequence from the *MYOD1* gene was used as a control of DNA input, as described elsewhere [44]. For absolute quantification, a standard curve was prepared by serial dilutions of a synthetic double-stranded *RPRML* DNA fragment (Integrated DNA Technologies, Coralville, IA, USA) starting at 1 ng/µL. A reference dilution was included in each plate for normalization between plates. Threshold cycle (Ct) values obtained from plasma samples were subsequently interpolated on the standard curve to determine the number of DNA copies/mL plasma using Rotor-Gene Q Series Software 2.3.5 (Qiagen, Hilden, Germany, RRID: SCR_015740). 

### 4.14. Statistical Analysis

Differences in *RPRML* expression between matched pairs of tumors and NTAM were evaluated using the Wilcoxon signed-rank test. To assess the difference in *RPRML* expression among clinicopathologic variables, IHC scores were treated as a continuous variable. Differences between two categories were evaluated using the Wilcoxon rank sum test (two-sided) or Welch’s unequal variances *t*-test. The Kruskal-Wallis test was applied if there were ≥3 categories. The effect of RPRML IHC score on overall survival (OS) was evaluated using univariate and multivariate Cox proportional hazards models adjusted by sex, age, and TNM (tumor-node-metastasis) stage. Kaplan-Meier survival analysis was performed as well and the overall comparison between curves was assessed using the log-rank test. Analyses were performed using the R software environment (RRID: SCR_001905). For in vitro functional assays, differences between groups were assessed by the Kruskal-Wallis and Dunn’s multiple comparison test using GraphPad Prism 8 (GraphPad Software, San Diego, CA, USA, RRID: SCR_002798). Three independent experiments were performed for each assay. To evaluate the ability of circulating methylated RPRML DNA to distinguish patients with GC from low-risk OLGA patient controls, receiver operating curve (ROC) analysis was performed using SPSS 15.0 (IBM, Armonk, NY, USA, RRID: SCR_002865). The best cut-off value was selected based on the maximization of the Youden Index [45]. In all cases, *p* < 0.05 was considered statistically significant.

## Figures and Tables

**Figure 1 ijms-21-09472-f001:**
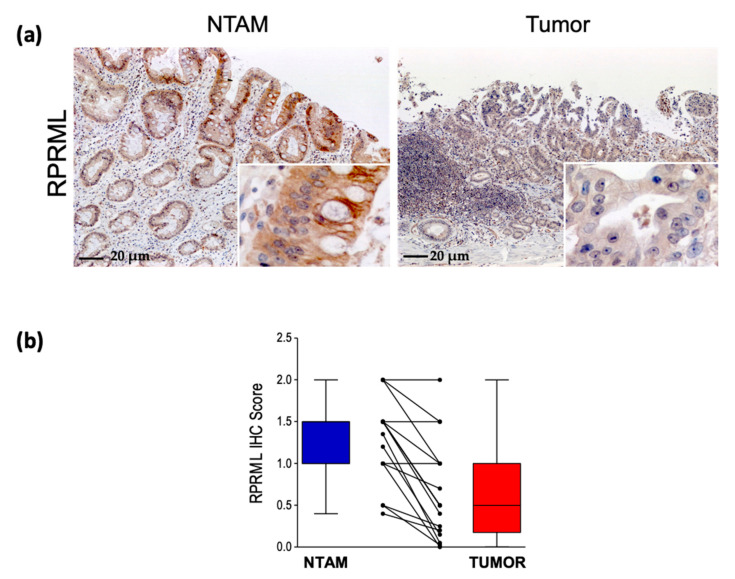
Reprimo-like (RPRML) expression is downregulated in gastric cancer (GC). (**a**) Representative images of RPRML immunohistochemical (IHC) staining assay in tumor and non-tumor adjacent mucosa (NTAM) samples from GC patients. Magnification: main image, ×100; insets, ×400. Scale bar = 20 μm. (**b**) Differential RPRML protein expression in 17 matched pairs of tumor and NTAM samples from GC patients. Box plots (median ± interquartile range (IQR)) and aligned dot plots of RPRML IHC scores are shown. Statistical analysis: Wilcoxon matched pairs signed rank test (*p* = 0.001).

**Figure 2 ijms-21-09472-f002:**
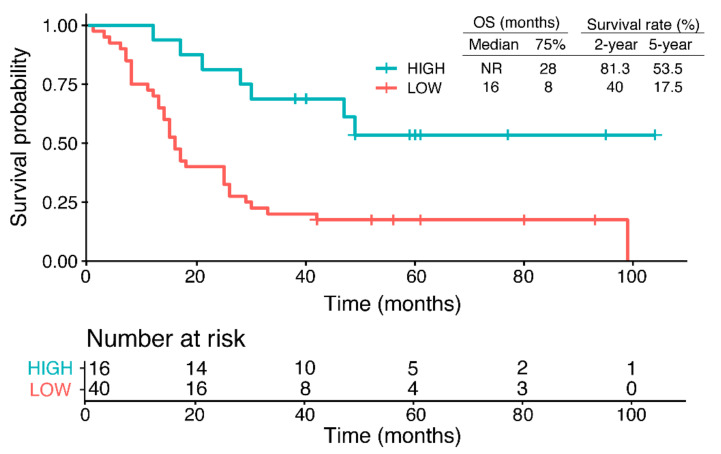
Overall survival (OS) according to the expression levels of RPRML in GC. Kaplan–Meier curves show the OS probability of patients with high versus low RPRML expression (cut-off IHC score = 0.162) among 56 advanced GC cases (TNM Stage III and IV). The median and 75% percentile OS times and 2- and 5-year survival rates are shown. The bottom panel shows the number of patients at risk.

**Figure 3 ijms-21-09472-f003:**
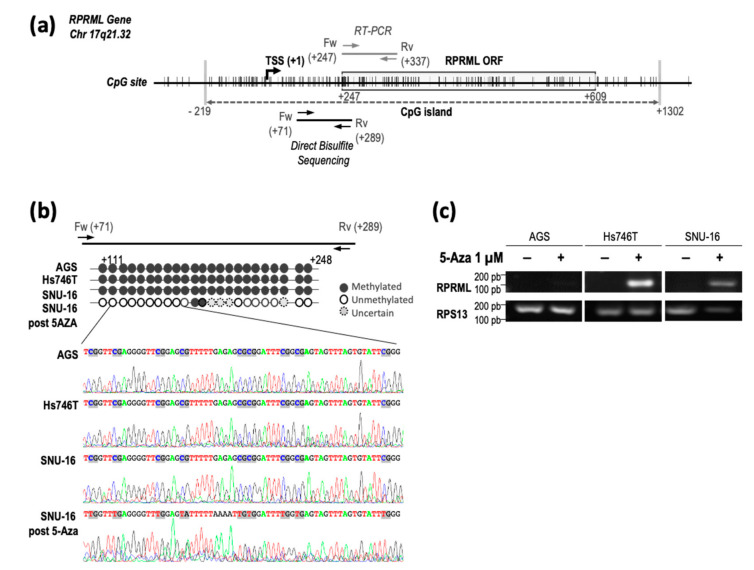
*RPRML* gene structure, methylation status, and transcript expression in GC cell lines after the 5-azacytidine (5-Aza) assay. (**a**) Schematic representation of *RPRML* gene structure, open reading frame (ORF), and CpG sites (vertical lines). A dense CpG island is depicted by a gray dashed arrow between nucleotides −219 and +1302 relative to the transcription start site (TSS). The region analyzed within the *RPRML* gene by direct bisulfite sequencing is located between nucleotides +71 to +289 relative to the TSS. RT-PCR primers are located between nucleotides +247 and +337. (**b**) Schematic representation and electropherograms of direct bisulfite sequencing results in the three GC cell lines. Individual CpG methylation status is represented by black (methylated) and white (unmethylated) circles. The bottom panel shows the result of direct bisulfite sequencing after treatment with 1 μM 5-Aza. (**c**) Re-expression of *RPRML* transcripts after treatment with 1 μM 5-Aza in three endogenously silent GC cell lines (AGS, Hs746T, SNU-16).

**Figure 4 ijms-21-09472-f004:**
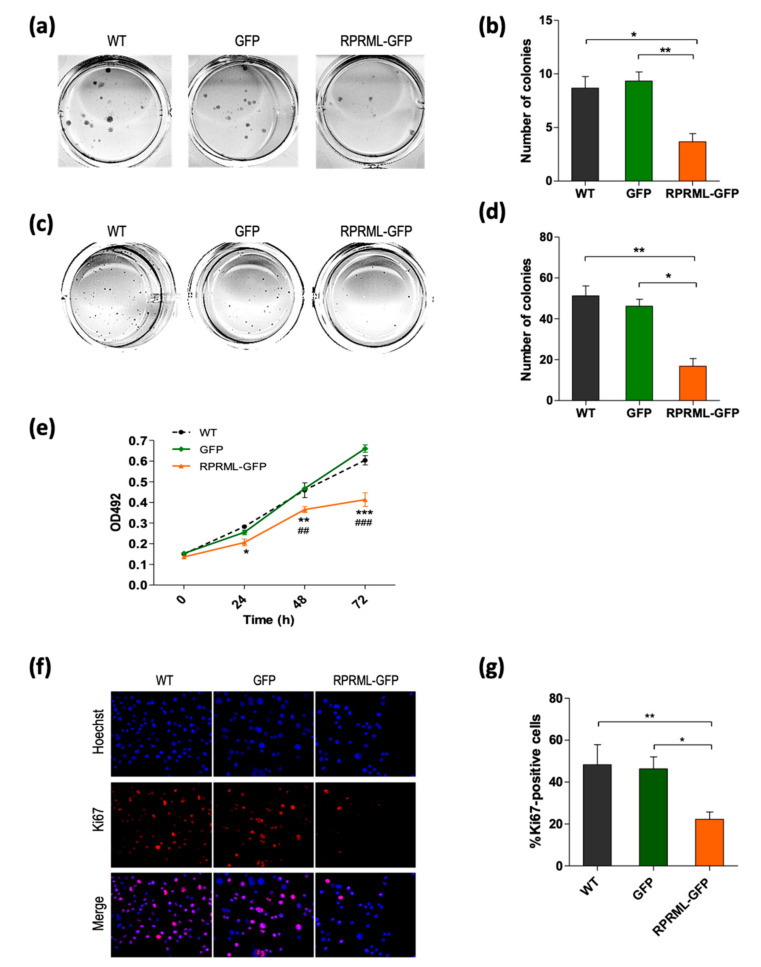
*RPRML* overexpression inhibits clonogenic capacity, anchorage-independent growth, and proliferation. (**a**) Representative image of a colony formation assay in wild type (WT), GFP-, and RPRML-GFP–overexpressing AGS cells. (**b**) Quantification of colony formation assay. The results are the mean number of colonies of three independent assays and the standard error of the mean (SEM) (bars). Statistical analysis: Kruskal-Wallis test (*p* = 0.007) followed by Dunn’s multiple comparison test (** *p* < 0.01, * *p* < 0.05). (**c**) Representative images of soft agar colony formation assay in WT, GFP-, or RPRML-GFP–overexpressing cells. (**d**) Quantification of soft agar colony formation assay. Colonies > 50 μm were counted. The results are the mean number of colonies of three independent assays and the SEM (bars). Statistical analysis: Kruskal-Wallis test (*p* = 0.001) followed by Dunn’s multiple comparison test (** *p* < 0.01, * *p* < 0.05). (**e**) Cell proliferation of WT, GFP-, and RPRML-GFP–expressing AGS cells was evaluated by MTS assay at 0 h, 24 h, 48 h, and 72 h (*n* = 3). Differences at each time point were evaluated by two-way repeated-measures ANOVA and a post hoc Bonferroni test. * WT vs. RPRML-GFP, ^#^ GFP vs. RPRML-GFP. *p* < 0.05 (*), *p* < 0.01 (**, ^##^), *p* < 0.0001 (***, ^###^). (**f**) Representative images of Ki67 immunofluorescence (red) and Hoechst staining (blue) in WT, GFP-, and RPRML-GFP–overexpressing AGS cells. (**g**) Percentage of Ki67-positive cells. Five random fields at ×10 magnification were quantified using ImageJ. Results represent the means of three independent experiments; bars indicate SEM. Differences between conditions were analyzed using the Kruskal-Wallis test followed by Dunn’s multiple comparison test (** *p* < 0.01, * *p* < 0.05).

**Figure 5 ijms-21-09472-f005:**
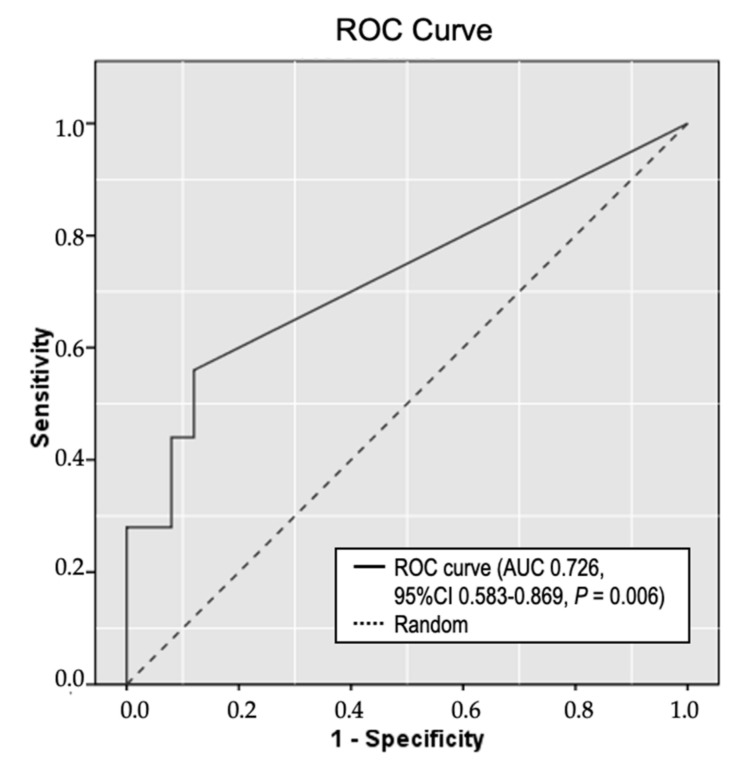
Receiver operating curve (ROC) analysis of circulating methylated *RPRML* DNA as a biomarker for detecting GC. Plasma samples from 25 GC cases and 25 age- and sex-balanced cancer-free controls were analyzed using the MethyLight assay. The solid line represents the ROC curve for circulating methylated *RPRML* DNA; the dashed line represents the line of no discrimination. The area under the curve (AUC) was 0.726 (95% confidence interval (CI): 0.583–0.869, *p* = 0.006).

**Table 1 ijms-21-09472-t001:** Association analysis of RPRML IHC score and overall survival by tumor–node–metastasis (TNM) stage subgroups.

	Subgroup	Hazard Ratio	*p*-Value
RPRML IHC Score	Stage I–II	3.26 (0.30–35.84)	0.334
	Stage III–IV	0.07 (0.01–0.46)	0.005

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
