# Peer review of "The Reprimo-Like Gene Is an Epigenetic-Mediated Tumor Suppressor and a Candidate Biomarker for the Non-Invasive Detection of Gastric Cancer"

_ijms, 2020, doi:10.3390/ijms21249472_

Round 1

Reviewer 1 Report

  • In the introduction section, except for mentioning the circulating tumor DNA (ctDNA), the authors can also mention the circulating tumor cells (CTC) as well as microRNA as potential candidates for either diagnostic or prognostic markers of GC. Besides, the authors have mentioned that the suppression of the tumor-suppressing genes contributes to the induction of GC but could you kindly provide some examples of such genes. A table with a tumor suppressor gene, chromosomal location, as well as the basic function might be beneficial to be added in the introduction just to provide the readers an overview in this area before reading about the RPRML gene specifically.
  • I would recommend modifying the discussion section a little bit by extracting from it two additional paragraphs - 1) Limitations of the study (as one paragraph) and what's important I would recommend adding a separate paragraph 'Conclusions' to extract the most relevant findings for the readers in just a few sentences.
  • I would like to congratulate the authors for such an interesting and novel work that might obviously have an impact on further understanding of the diagnosis of gastric cancer.

Author Response

Comments from Reviewer 1:

Comment 1: In the introduction section, except for mentioning the circulating tumor DNA (ctDNA), the authors can also mention the circulating tumor cells (CTC) as well as microRNA as potential candidates for either diagnostic or prognostic markers of GC.

Thank you for pointing this out. We agree with this comment. Therefore, we have mentioned other candidate biomarkers in the introduction section, lines 61-66, page 2:

“Moreover, it has been demonstrated that circulating tumor-specific methylated DNA can be detected in different biological fluids, as well as microRNAs, long non-coding RNAs, extracellular vesicles, and circulating tumor cells (CTCs) [9]. Thus, epigenetic-mediated TSG silencing not only contributes to the pathogenesis of GC but has also been proposed as a candidate biomarker for the non-invasive diagnosis and monitoring of disease [10-12].”

Comment 2: Besides, the authors have mentioned that the suppression of the tumor-suppressing genes contributes to the induction of GC but could you kindly provide some examples of such genes. A table with a tumor suppressor gene, chromosomal location, as well as the basic function might be beneficial to be added in the introduction just to provide the readers an overview in this area before reading about the RPRML gene specifically.

We agree that this suggestion will be beneficial to contextualize tumor suppressor genes silenced by DNA methylation in GC. However, we believe it would be more appropriate to incorporate a paragraph rather than a table since there are too many examples of such genes, which have been comprehensively reviewed in recent articles. We have added the suggested content in the following paragraph in the introduction section, lines 53-61, page 2:

“Comprehensive studies have revealed that extensive epigenetic DNA hypermethylation on tumor suppressor genes (TSG) is a recurrent phenomenon in GC [4]. These aberrant patterns of DNA methylation arise early during carcinogenesis, and when occurring within the nearest region surrounding the transcription start site (TSS ±200 bp) may lead to TSG silencing [5,6]. These genes are involved in crucial tumorigenic processes such as cell cycle regulation (p16NK4a, p15INK4b, p14ARF, RPRM, RUNX3), apoptosis (XIAP, BCL2, DAPK, BNIP3, TMS1, CASP8, GPX3, RNF180), DNA repair (hMLH 1, MSH2, MGMT), migration and invasion (CDH1, CDH4, PCDH10, RASSF1A, ZIC1, APC, GRIK2, FLNC, LOX, TIMP3, TSP1), among others, and have been associated with the progression and poor prognosis of GC [7,8]. “

Comment 3: I would recommend modifying the discussion section a little bit by extracting from it two additional paragraphs - 1) Limitations of the study (as one paragraph) and what's important I would recommend adding a separate paragraph 'Conclusions' to extract the most relevant findings for the readers in just a few sentences.

Thank you for this recommendation. We have added separated paragraphs for limitations and conclusions. You can find the following modification in the last two paragraphs of the discussion section, lines 278-296, page 10:

“Our findings are subject to certain limitations. It should be noted that due to the exploratory nature of this study, a small case-control design was used to evaluate the potentiality of circulating methylated RPRML DNA as a non-invasive biomarker of GC. We did not consider variables such as H. pylori and Epstein Barr virus infections which have been reported to influence the methylation status in the stomach [35]. In addition, the diagnostic accuracy of circulating methylated RPRML DNA and the prognostic value of RPRML protein expression warrant validation in independent cohorts. Finally, further functional analyses and animal studies will be required to fully validate the role of RPRML and its specific signaling pathways in GC.

Despite these considerations, this study constitutes a first step toward a broader characterization of this hitherto undescribed gene in human biology and pathology. Our results suggest that RPRML is a TSG, downregulated by DNA methylation in GC, and that circulating methylated RPRML DNA can discriminate patients with GC from cancer-free controls. Thus, these findings provide justification for larger clinical studies to further assess its value in multi-biomarker panel approaches for non-invasive diagnosis of GC.”

I would like to congratulate the authors for such an interesting and novel work that might obviously have an impact on further understanding of the diagnosis of gastric cancer.

 Thank you.

Reviewer 2 Report

This is an interesting and well performed study of a novel gene  (REprimo-like gene). The same authors have a long experience in Reprimo gene, reprimo-like gene showed a limited number of studies in the past.

Only minor comments:

1) added a comment in the discussion regarding the only gene found associated caspase 3.

2) age limit score will be preferable of 50 years rather than 60 years. Could you revise data accordantly or at least in 3 categories to evidence a possibile tendency ??

3) H. bacter pylori is a factor inportant for GC and for methylation. Could you included this information as an important covariable ??

4) Added a comparison between reprimo and reprimo-like methylation data from peripheral blood. What is the potential better marker for GC prediction and/or OS between them ??

Author Response

Comments from Reviewer 2:

This is an interesting and well performed study of a novel gene  (Reprimo-like gene). The same authors have a long experience in Reprimo gene, reprimo-like gene showed a limited number of studies in the past.

Thank you for this comment.

Only minor comments:

Comment 1: added a comment in the discussion regarding the only gene found associated caspase 3.

Thank you for this suggestion and for highlighting this point. We have made substantial modifications to the paragraph referring to clinical associations in the discussion section and added a comment on the association of RPRML expression with cleaved caspase-3 on lines 264-277 of the manuscript, page 10:

“Correspondingly, analysis of RPRML expression in NTAM and tumor tissues showed consistent downregulation in clinical samples and associated with worse prognosis in advanced stages of GC. These results suggest that the loss of RPRML protein expression is an independent prognostic factor and may drive GC progression. Thus, it provides the opportunity for exploring the potentiality of RPRML as an actionable target for advanced GC, as has been previously proposed for its homolog RPRM using CRISPR technology [33]. Interestingly, the absence of RPRM gene expression in clinical samples is not associated with poor prognosis in GC patients [15]. However, the apoptosis-resistant phenotype mediated by Survivin, a member of the inhibitor-of-apoptosis protein (IAP) family, has been reported to be detrimental to survival in GC only in the absence of RPRM gene expression [15]. In addition, the loss of RPRML protein expression in clinical samples was associated with reduced Cl-caspase-3 immunostaining. This finding suggests that RPRML may have a role in resisting apoptosis, which has been previously associated with poor GC prognosis [34].”

Comment 2: age limit score will be preferable of 50 years rather than 60 years. Could you revise data accordantly or at least in 3 categories to evidence a possible tendency ??

We agree with this suggestion. Accordingly, we have changed the age limit score to 3 categories: ≤50, 51-65, and >65 as you will find in Supplementary Table I of the revised manuscript. However, we still did not find any significant difference in RPRML expression.

Characteristic

N

%

RPRML IHC Score

Median

P Value

Age (years)

0.83c

   ≤50

13

14.61

0.013

   51-65

41

46.07

0.013

   >65

35

39.32

0.038

Comment 3: H. pylori is a factor important for GC and for methylation. Could you include this information as an important covariable ??

Thank you for this suggestion. It would have been interesting to explore this aspect. Unfortunately, we do not have H. pylori information for these patients, and given the pandemic situation we cannot assess this data at this time. However, we included a comment on this limitation in the discussion section, lines 281-282, page 10:

 “We did not consider variables such as H. pylori and Epstein Barr virus infections which have been reported to influence the methylation status in the stomach [35].”

Comment 4: Added a comparison between reprimo and reprimo-like methylation data from peripheral blood. What is the potential better marker for GC prediction and/or OS between them ??

Thank you for this recommendation. We have added a comparison with previously reported performance (odds ratio) of RPRM on line 237-243, in the discussion section, page 9:

“A recent meta-analysis compared the clinical performance of methylation-based blood biomarkers for detecting GC [26]. Among biomarkers with significant discriminatory capacity, RPRM, the homolog of RPRML, was proposed as one of the most promising candidates. Our results of the odds ratio of RPRML (9.34, 95% CI: 2.20–39.46) fall among the mean odds ratio of these reported biomarkers, which ranged from 3.16 (95% CI: 1.47–6.81) for MGMT to 111.1 (95% CI: 36.67–336.59) for RPRM [26]. However, a wide 95% CI was observed in each of these candidates, indicating low-precision accuracy and inconsistency between aggregated studies.”